# Widespread nociceptive maps in the human neonatal somatosensory cortex

**Laura Jones[1]\*, Madeleine Verriotis[1]†, Robert J Cooper[2],
Maria Pureza Laudiano-Dray[1], Mohammed Rupawala[1], Judith Meek[3],
Lorenzo Fabrizi[1], Maria Fitzgerald[1]\***

[1]Department of Neuroscience, Physiology & Pharmacology, University College London, London, United Kingdom; [2]Department of Developmental Neuroscience, University College London, London, United Kingdom; [3]Elizabeth Garrett Anderson Obstetric Wing, University College London Hospitals, London, United Kingdom

**Abstract** Topographic cortical maps are essential for spatial localisation of sensory stimulation and generation of appropriate task-related motor responses. Somatosensation and nociception are finely mapped and aligned in the adult somatosensory (S1) cortex, but in infancy, when pain behaviour is disorganised and poorly directed, nociceptive maps may be less refined. We compared the topographic pattern of S1 activation following noxious (clinically required heel lance) and innocuous (touch) mechanical stimulation of the same skin region in newborn infants ($n$ = 32) using multi-optode functional near-infrared spectroscopy (fNIRS). Within S1 cortex, touch and lance of the heel elicit localised, partially overlapping increases in oxygenated haemoglobin concentration ($\Delta$[HbO]), but while touch activation was restricted to the heel area, lance activation extended into cortical hand regions. The data reveals a widespread cortical nociceptive map in infant S1, consistent with their poorly directed pain behaviour.

**\*For correspondence:**
laura.a.jones@ucl.ac.uk (LJ);
m.fitzgerald@ucl.ac.uk (MF)

**Present address:** †Department of Developmental Neuroscience, University College London, Great Ormond Street Institute of Child Health, London, United Kingdom

**Competing interest:** The authors declare that no competing interests exist.

## Editor's evaluation

This paper is of interest to developmental neuroscientists who study the early stages of cortical maturation and specialization, particularly in the context of somatosensory and pain system development. The authors suggest that, relative to the infant touch somatotopic map, the infant nociceptive map is more widespread and poorly localised, consistent with infants' poorly directed pain behaviour.

## Introduction

Somatotopically organised cortical maps of activity evoked by innocuous or noxious mechanical stimulation allow us to localise our sense of touch or pain (*Penfield and Boldrey, 1937*; *Harding-Forrester and Feldman, 2018*), and may also convey computational advantages in the relay of afferent information to higher brain areas (*Thivierge and Marcus, 2007*). In adults, overlapping regions are involved in the cortical processing of noxious and innocuous mechanical stimulation (*Kenshalo et al., 2000*; *Lui et al., 2008*) and detailed fMRI analysis reveals a fine-grained somatotopy for nociceptive inputs in primary somatosensory (S1) cortex that are aligned with activation maps following innocuous tactile stimuli, suggesting comparable cortical representations for mechanoreceptive and nociceptive signals (*Mancini et al., 2012*).

A whole-body topographical map of innocuous mechanical stimulation develops in the sensorimotor cortices over the early postnatal period in rats, which represent the human final gestational trimester (*Seelke et al., 2012*). Distinct representations of the hands and feet can be observed from

31 weeks using fMRI (*Dall'Orso et al., 2018*), becoming increasingly localised by term age (*Allievi et al., 2016*). While haemodynamic responses to a clinically required heel lance have been recorded from 28 weeks using functional near-infrared spectroscopy (fNIRS) (*Slater et al., 2006*) and can be distinguished from innocuous tactile evoked brain activity in EEG recordings from 34 to 35 weeks (*Fabrizi et al., 2011*), the source of this activity and topographic representation of these two modalities have not been mapped, or their alignment established, in the infant cortex.

Infant pain behaviour is exaggerated and disorganised in newborn rodents and human infants (*Fitzgerald, 2005*; *Fitzgerald, 2015*; *Cornelissen et al., 2013*). Poor spatial tuning of nociceptive reflexes and receptive fields is a feature of the developing somatosensory system, followed by the emergence of adult organisation through activity-dependent refinement of synaptic connections (*Beggs et al., 2002*; *Schouenborg, 2008*; *Koch and Fitzgerald, 2013*). We hypothesised that this developmental process is reflected in ascending nociceptive signals to S1, leading to widespread S1 activation and poor spatial localisation of noxious events in early life.

To test this hypothesis, we used multioptode fNIRS to map nociceptive and innocuous mechano-receptive activity across the infant sensorimotor cortex. fNIRS is a non-invasive measure of cerebral haemodynamic changes which can be performed at the bedside, while in skin-to-skin holding and in a naturalistic hospital setting during clinically required procedures. Using the temporal and spatial profiles of haemodynamic responses to a noxious heel lance and an innocuous touch of the hand and the heel, we show that haemodynamic activity elicited by noxious and innocuous mechanical stimuli have partially overlapping topographies in the human infant S1 cortex but that the two maps are not aligned. Noxious stimulation of the heel in the newborn evokes more widespread S1 activation than innocuous stimulation, that extends into inferior regions, normally associated with representation of the hand.

## Results

### Innocuous hand and heel touch evoked activity is somatotopically organised in the newborn infant S1 cortex

We first established the cortical topography of touch activation in newborn infants by mapping the extent of activation in the contralateral somatosensory (S1) cortex following innocuous mechanical stimulation (touch) of the hand and heel. *Figure 1a and b* shows a significant and localised increase in average concentration of oxygenated haemoglobin ($\Delta$[HbO]) in contralateral optode channels following touch of each body area ($n = 11$, hand touch; $n = 16$ heel touch). Touch stimulation of the hand elicited significant increases in seven channels, with a maximum change (0.33 µM at 9.2 s post-stimulus) at the channel corresponding to the FCC5 position of the 10:5 placement system (*Oostenveld and Praamstra, 2001*), while touch of the heel elicited significant increases in seven channels, with a maximum change (0.35 µM at 15.8 s) at the channel corresponding to the CPP1 10:5 position.

The somatotopically localised increases in $\Delta$[HbO] were accompanied by a widespread decrease in $\Delta$[Hb] over the whole peri-rolandic area (hand: significant decreases in eight channels [peak change: –0.18 µM at 17.6 s]; heel: significant decreases in 14 channels [peak change: –0.21 µM at 11.1 s]). An inverse response (significant decrease in $\Delta$[HbO], significant increase in $\Delta$[Hb], or both) was mostly restricted to channels surrounding the hand and foot areas of the S1, respectively. Individual channel data is shown in *Figure 1—source data 1*.

Image reconstruction of the channel data (*Figure 2a and b*) shows that the topography of touch peak activation in the newborn infant S1 is consistent with the known adult S1 topography: the area representing the foot lies in the superomedial postcentral gyrus, while the area for the hand is more lateral and inferior (*Penfield and Boldrey, 1937*; *Harding-Forrester and Feldman, 2018*; *Willoughby et al., 2020*).

### Noxious lance of the heel elicits widespread activation extending into inferior S1

We next mapped activation in the contralateral S1 following a noxious, clinically required, lance stimulus to the heel in newborn infants. The average channel response (*Figure 1c*) and the image reconstruction (*Figure 2c*) show the significant and widespread increase in $\Delta$[HbO] following the heel lance,

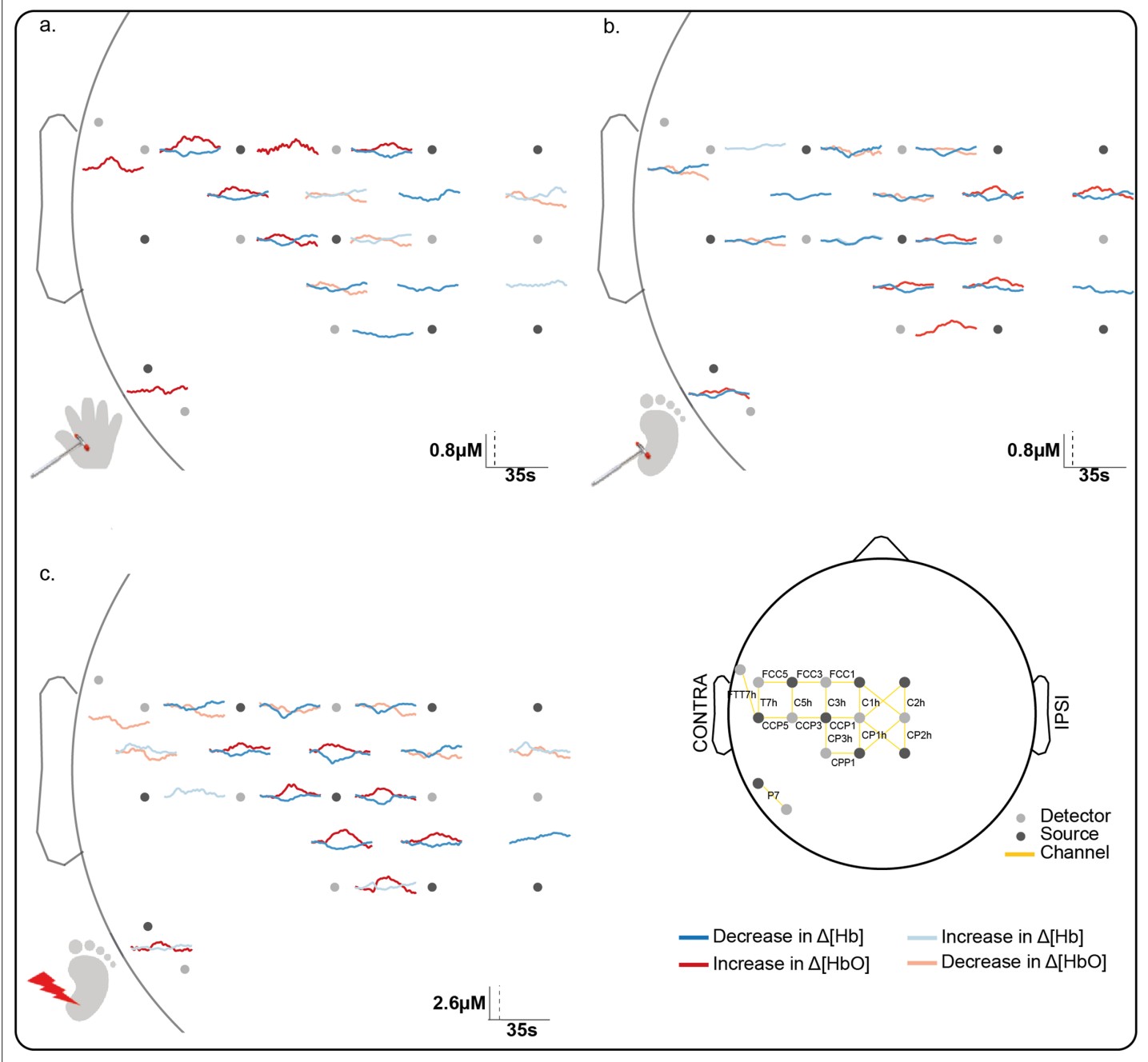

**Figure 1.** Significant channel-wise haemodynamic response following innocuous (touch) and noxious (lance) mechanical stimulation of hand and heel. Average significant change in concentration of oxygenated (Δ[HbO]) (red) and deoxygenated haemoglobin (Δ[Hb]) (blue) during (**a**) hand touch (*n* = 11), (**b**) heel touch (*n* = 16), and (**c**) heel lance (*n* = 11). Channels with significant increases in Δ[HbO] and decreases in Δ[Hb] (i.e. canonical response) during the activation period are shown with solid dark lines, channels with an inverse response only (decrease in Δ[HbO] and increase in Δ[Hb]) are shown with pale solid lines. Note the difference in the scale bar between touch and lance. The equivalent plots for non-significant changes are shown in *Figure 1—figure supplement 1* and details of individual channel responses are in *Figure 1—source data 1*.

The online version of this article includes the following source data and figure supplement(s) for figure 1:

**Source data 1.** Significant concentration changes at each channel following innocuous mechanical stimulation (touch) of the heel and the hand and following heel lance.

**Figure supplement 1.** Non-significant channel-wise haemodynamic responses following innocuous and noxious mechanical stimulation of hand and heel.

**Figure supplement 2.** Average channel-wise Δ[HbO] response following innocuous mechanical stimulation of the hand and heel with and without excluding trials with movement.

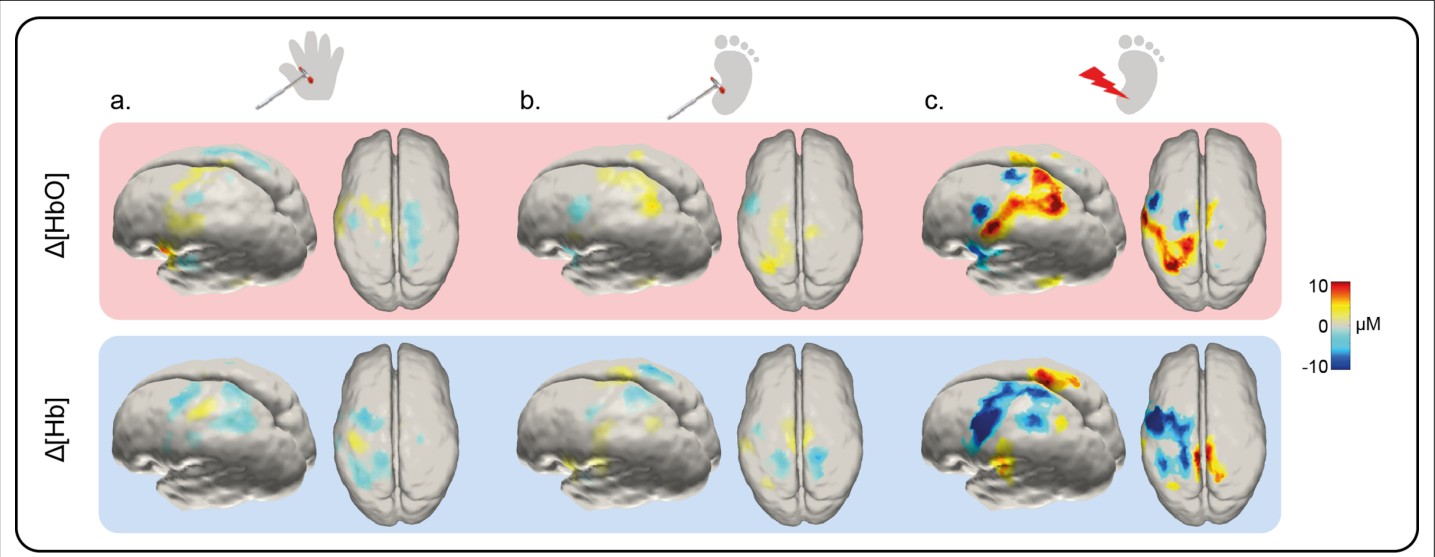

**Figure 2.** Image reconstruction at peak latency of the Δ[HbO] and Δ[Hb] response to an innocuous (touch) and noxious (lance) mechanical stimulation of hand and heel. Significant changes (compared to baseline) in concentration of oxygenated (Δ[HbO]) (top row) and deoxygenated haemoglobin (Δ[Hb]) (bottom row) following (**a**) hand touch (n = 11), (**b**) heel touch (n = 16), and (**c**) heel lance (n = 11).

which extends beyond the somatotopic area for heel touch to encompass inferior areas of S1, which were associated with touch of the hand.

Heel lance elicited significant increases in Δ[HbO] in eight channels, with a maximum increase (0.96 μM at 14.5 s) at the channel corresponding to the CP3h 10:5 position. Four of the channels with a significant increase in Δ[HbO] following the lance also had a significant increase in Δ[HbO] following touch of the heel. Notably two channels also displayed a significant increase following touch of the hand, demonstrating a topographic overlap with both the hand and heel touch responses. The accompanying decrease in concentration of deoxygenated haemoglobin (Δ[Hb]) was widespread (significant decreases in 11 channels; peak change 1.03 μM at 9.1 s), and an inverse response was found in all channels surrounding those with a canonical response (*Figures 1c and 2c*).

## Newborn infant nociceptive maps are more widespread and not somatotopically aligned with innocuous mechanoreceptive maps

We then compared the responses to heel touch and lance in terms of amplitude, latency and position of peak change, and extent and overlap of the overall areas of activation at peak latency. Heel touch and lance elicited a peak increase in Δ[HbO] at the same latency and location (distance between peaks = 1.65 mm, p = 0.156; difference in peak latency = 2 s [14 vs. 16 s], p = 0.358; *Figure 3a–b*), however this was significantly larger (maximum Δ[HbO]: 16.06 vs. 5.43 μM, p = 0.001) and wider (FWHM area: 51.41 vs. 43.79 mm², p = 0.019) following lance stimulation. The overall area of activation was also significantly more widespread following heel lance (Δ[HbO] overall area: 455.27 vs. 259.51 mm², p = 0.009) and only partially overlapped to that following touch (*Figure 3a*). The increase in Δ[HbO] was significantly larger following lance compared to touch across 50% of the lance-only activation area (*Figure 3a* – red area, *Figure 3c*) and across 26% of the overlap area (*Figure 3a* – orange area, *Figure 3c*). However, there was no significant difference in Δ[HbO] changes between lance and touch across the touch-only activation area (*Figure 3a* – pink area). This shows that the significantly larger and wider peak response and more widespread overall area of activation following lance is not due to a greater systemic haemodynamic change following this stimulus.

The location, but not latency, of the peak change in Δ[Hb] was significantly different following heel touch and lance (distance between peaks = 51.88 mm, p = 0.041; difference in peak latency = –19.3 s [9.5 vs. 28 s], p = 0.028), which was also significantly larger (maximum Δ[Hb]: –22.54 to –6.62 μM, p < 0.001) and wider (Δ [Hb] FWHM area: 15.31 vs. 4.63 mm², p < 0.001) following heel lance (*Figure 3— figure supplement 1*). The overall area of activation was also significantly more widespread following

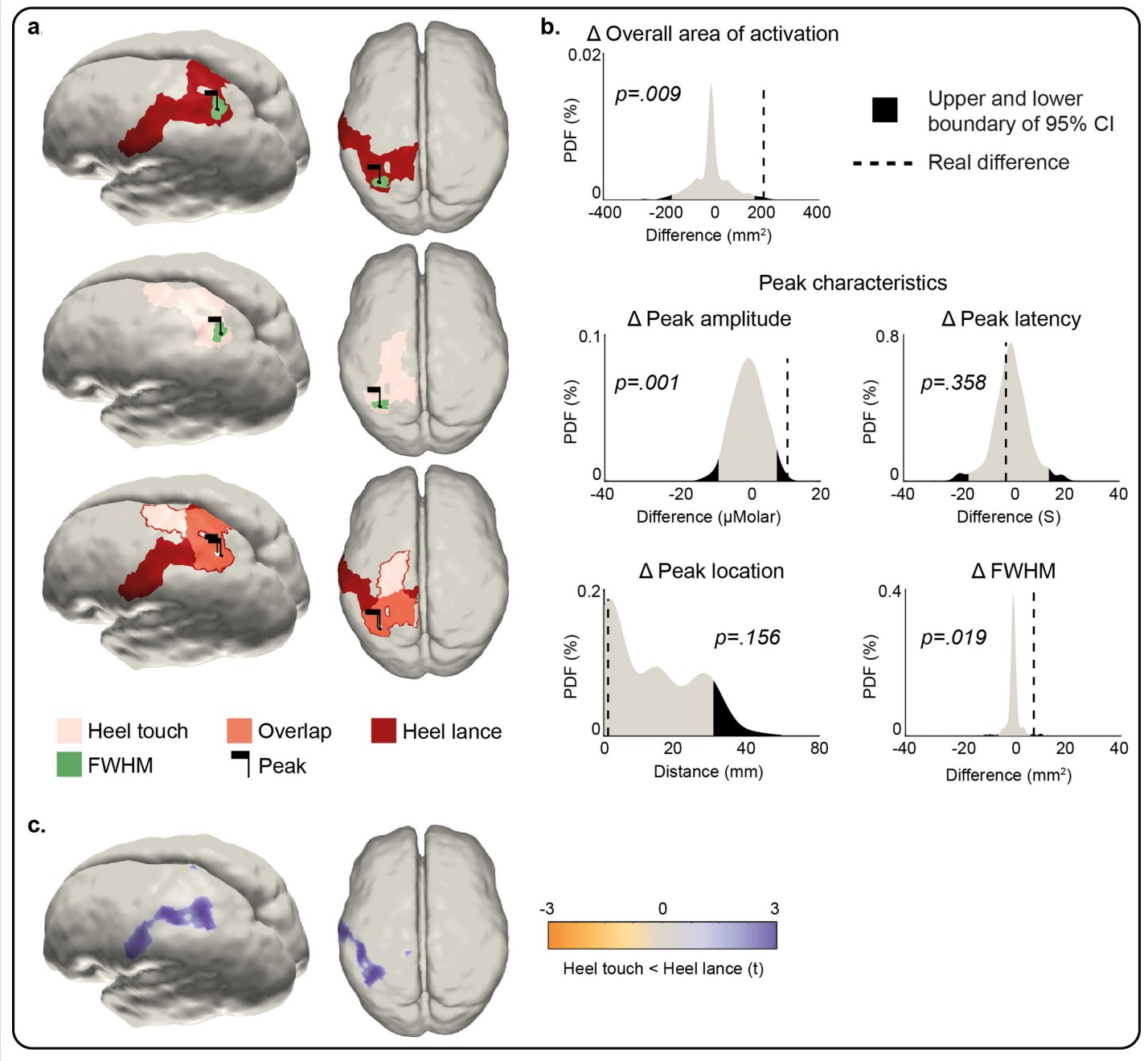

**Figure 3.** Comparison of the peak and area of activation of the Δ[HbO] response to an innocuous (touch) and noxious (lance) mechanical stimulation of the heel. (**a**) Overall area of significant changes in concentration of oxygenated haemoglobin (Δ[HbO]) following heel lance (red), heel touch (pink), and both (orange). Black flags demark the location of peak changes and green areas the extent of their full-width half-maximum (FWHM). (**b**) Statistical position of real differences between heel touch and lance in peak amplitude, FWHM, latency and location, and overall area of activation in respect to non-parametric null distributions obtained with bootstrapping and phase scrambling. (**c**) Comparison of response magnitude at each node. Results show the t-statistic at significantly different nodes within the areas of activation shown in (**a**). The equivalent plots for Δ[Hb] are shown in *Figure 3— figure supplement 1* and *Figure 3—source data 1*.

The online version of this article includes the following source data and figure supplement(s) for figure 3:

**Source data 1.** Comparison of Δ[HbO] and Δ[Hb] responses to an innocuous (touch) and noxious (lance) mechanical stimulation of the heel.

**Figure supplement 1.** Comparison of the peak and area of activation of the Δ[Hb] response to an innocuous (touch) and noxious (lance) mechanical stimulation of the heel.

heel lance (Δ[Hb] overall area: 475.24 vs. 95.53 mm², p < 0.001). The key difference between the two patterns of activation is that the heel touch Δ[HbO] response is limited to the areas of the S1/M1 associated with the foot, whereas the lance Δ[HbO] response extends towards other more ventral regions of S1.

## Discussion

### A widespread nociceptive topographic map in infant S1 that overlaps but is not aligned to the innocuous mechanoreceptive map

Somatosensory maps of cortical activity evoked by a cutaneous tactile or noxious stimulus provide a framework for localising the sense of touch or pain (*Treede et al., 1999*; *Thivierge and Marcus, 2007*). The adult primate S1 has a defined somatotopic organisation of tactile and nociceptive cortical receptive fields (*Andersson et al., 1997*; *Kenshalo et al., 2000*) including spatially precise cortical maps of Aδ and Aβ afferent fibre input (*Chen et al., 2011*). Human fMRI studies show that adult somatotopic maps of noxious and non-noxious mechanical stimulation substantially overlap (*Lui et al., 2008*) and detailed analysis reveals a fine-grained somatotopy for nociceptive inputs in S1 cortex that are highly aligned with maps of innocuous tactile stimuli, suggesting comparable cortical representations for mechanoreceptive and nociceptive signals (*Mancini et al., 2012*). Here, we have shown that this comparable representation is not present in the newborn infant S1 cortex.

Noxious mechanical stimulation evokes a larger peak increase in concentration of oxygenated haemoglobin (Δ[HbO]) and decrease in concentration of deoxygenated haemoglobin (Δ[Hb]) compared to innocuous stimulation of the same body area at comparable location and latency as reported elsewhere (*Bartocci et al., 2006*; *Slater et al., 2006*; *Verriotis et al., 2016b*). The fact that the noxious activation is greater than the touch evoked activity is presumably due to hyperaemia associated with greater depolarisation and spike activity within the activated areas. However, it does not explain the differing topography of the overall area of activation reported here. Although a lance, being a more intense stimulus, could be thought to cause more widespread cortical activation throughout the cortex, here the response to the lance is only significantly larger than touch in 50% of lance-only area of activation, 26% of the lance and touch overlap area, and nowhere in the touch-only activation area. Our results instead suggest that the response to a noxious and innocuous mechanical stimulus is not spatially aligned. Within S1 itself, the infant has a distinct somatotopic map for touch, similar to that described in adults, with the area representing the foot lying in the superomedial postcentral gyrus, and the area for the hand located more inferiorly (*Penfield and Boldrey, 1937*; *Blake et al., 2002*; *Akselrod et al., 2017*), consistent with previous reports in newborn infants (*Dall'Orso et al., 2018*). Noxious heel lance, on the other hand, evokes widespread activity within S1, peaking in a similar area of the superomedial postcentral gyrus as touch activity, but extending to the hand representation area. All stimuli elicited a significant response in the control channel which may suggest that the responses extended more posteriorly to S1, nevertheless, our data shows that the S1 somatotopic nociceptive map is not as precise as the touch map in the newborn.

It should be noted that fNIRS is an indirect measure of neuronal activation and may be confounded by systemic vascular changes such as increases in blood pressure, which may be more likely following a noxious vs. innocuous stimulus. However, we removed global changes common across channels using principal component analysis (PCA), and our results point to misaligned noxious and innocuous activity, with widespread noxious activity across the S1 specifically, but not the whole cortex as one would expect following systemic changes. Several channels/nodes did not show a significant response following the heel lance compared to baseline, and the heel lance did not have a greater response compared to heel touch at every node. Together, these findings suggest that the greater magnitude of the heel lance response is not due to systemic changes elicited by the noxious stimulus. However, as fNIRS (as with fMRI) is a measure of the vascular response following neuronal activity, the precise degree of localisation of either neuronal response is limited.

Measuring the cortical haemodynamic response to innocuous and noxious mechanical stimulation in neonates fNIRS is ideally suited to a study of this kind as recording and sensory stimulation, including clinically required heel lance, can be performed at the infant cotside (*Bartocci et al., 2006*; *Slater et al., 2010*; *Kashou et al., 2017*; *Verriotis et al., 2016b*). Other methods of measuring this either do not provide sufficient spatial information and source localisation, such as EEG recording of

nociceptive-related ERPs (*Fabrizi et al., 2011*; *Jones et al., 2018*) or are limited by the use of experimental 'pinprick' stimulators, that for ethical reasons are not actually painful, such as in fMRI studies (*Goksan et al., 2015*).

The change in the Δ[HbO] and Δ[Hb] following sensory stimulation is an indirect measure of neural activity: simultaneous vertex EEG and fNIRS recordings over S1 show that haemodynamic and neural responses are related in magnitude (*Verriotis et al., 2016b*). Following all stimuli, and consistent with the mature canonical response, channels showing a significant increase in Δ[HbO] also had a smaller decrease in Δ[Hb]. Regional overperfusion following neuronal activation, beyond that required by metabolic demands, means that less Hb is removed from the region compared to the oversupply of HbO. However, the decrease in Δ[Hb] was more widespread (but smaller in magnitude) compared to the localised increase in Δ[HbO]. This type of response, not previously reported in infants (*de Roever et al., 2018*), suggests that more blood is leaving the region (removing Hb) compared to the incoming supply (no significant change in Δ[HbO] in peripheral channels) due to immature regulation of cerebral blood flow (CBF). There are multiple mechanisms by which blood vessels dilate and CBF increases following neural activation, including arterial $CO_2$ and $O_2$ concentrations, which relax/contract the smooth muscle cells of cerebral arteries and arterioles (*Kety and Schmidt, 1948*), and astrocyte and pericyte activity which controls vessel diameter and the propagation of vasodilation along the vascular tree (*Takano et al., 2006*; *Cai et al., 2018*), many of which are still developing in the newborn (*Pryds and Greisen, 1989*; *Binmöller and Müller, 1992*; *Fujimoto, 1995*) leading to rapid changes in CBF over the first postnatal days as cerebral circulation adapts (*Meek et al., 1998*).

The infants in this study were held skin to skin, swaddled in their mother's arms in a naturalistic setting, which is a major advantage of fNIRS recording over fMRI for human developmental studies of brain function. Video recording and investigator scoring confirmed that while some infant movement and maternal touching did take place and that some babies did move following the lance, these movements spanned different body parts and latencies such that any associated cortical response would be cancelled during the averaging process. Indeed, comparison of the time-series following heel touch with and without removing trials with known movement showed no significant difference (*Figure 1—figure supplement 2*). Furthermore, 33–50% of babies did grimace for up to 7 s following the lance, but if these facial movements mediated the response following the lance, this would have prolonged the peak or duration of the change in [HbO], while in fact the latency and time course of the response to both stimuli was the same (see *Figure 1*).

## Differential development of somatosensory and nociceptive topographic maps

A whole-body topographical map of innocuous mechanical stimulation develops in the sensorimotor cortices over the early postnatal period in rats (*Seelke et al., 2012*), which represents the human final gestational trimester. In humans, distinct representations of the hands and feet can be observed from 31 weeks' gestation, using fMRI (*Dall'Orso et al., 2018*), and from 28 weeks using neural activity recorded from the scalp (*Donadio et al., 2018*; *Whitehead et al., 2018*; *Whitehead et al., 2019*). The response to innocuous mechanical stimulation was more localised in S1 than the wider and less refined topographical map of noxious mechanical stimulation, suggesting a slower maturation of the S1 circuitry involved in nociceptive processing compared to touch processing in the infant brain.

In rodents, at every level of the developing somatosensory central nervous system, tactile processing matures before nociceptive processing (*Fitzgerald, 2005*; *Koch and Fitzgerald, 2013*; *Chang et al., 2016*; *Chang et al., 2020*; *Verriotis et al., 2016a*) consistent with a delayed refinement of a cortical nociceptive map. Widespread nociceptive cortical maps are consistent with infant pain behaviour, characterised by exaggerated and disorganised nociceptive reflexes in both rodent pups and human neonates (*Fitzgerald, 2005*; *Fitzgerald, 2015*), and which can fail to remove a body part from the source of pain (*Waldenström et al., 2003*). Nociceptive reflexes following noxious heel lance are larger in magnitude and significantly more prolonged in human infants compared to adults (*Cornelissen et al., 2013*) and have widespread cutaneous receptive fields that encompass the whole lower limb (*Andrews and Fitzgerald, 1994*). This lack of organisation could be reflected in the ascending spinothalamic and thalamocortical projections, delaying the maturation of S1 cortical nociceptive maps in the newborn. Topographic maps are established and aligned via multiple mechanisms, including molecular cues, spontaneous or sensory-dependent remodelling, and refinement.

**Table 1.** Infant demographics.
Demographic information about the subjects that received tactile and noxious stimuli of heel and hand.

| | Heel lance | Heel touch | Hand touch | p |
|---|---|---|---|---|
| N | 11 | 16 | 11 | |
| GA (weeks$^{+days}$) | $39^{+2}$ ($35^{+2}$–$41^{+5}$) | $39^{+4}$ ($35$–$42^{+3}$) | $39^{+2}$ ($37^{+5}$–$41^{+3}$) | 0.287 |
| PNA (days) | 4 (0–7) | 3 (0–6) | 3 (0–4) | 0.115 |
| Females | 4 (36%) | 6 (38%) | 5 (45%) | 0.889 |
| Birth weight (g) | 3134 (2220–4072) | 3250 (2360–4080) | 3300 (2450–3754) | 0.774 |
| Caesarean deliveries | 2 (18%) | 8 (50%) | 3 (27%) | 0.196 |
| Head circumference (cm) | 34 (32–35.5) | 34.25 (31–37) | 34 (32.5–36) | 0.900 |

Values represent median and range or proportion. GA = gestational age (weeks from the first day of the mother's last menstrual cycle to birth); PNA = postnatal age (days since birth). No significant difference was found in any demographic parameter across the three groups (one-way ANOVA results in the last column).

Initially, somatosensory maps are diffuse and overlapping, but in the rodent somatosensory cortex, excitatory thalamocortical afferents undergo activity-dependent refinement to sharpen these maps (*Iwasato et al., 1997*). Equally important is the maturation of inhibitory interneuron sensory maps which, in contrast, expand over development in an experience-dependent manner (*Quast et al., 2017*). Slow developmental broadening of an inhibitory nociceptive network may explain the widespread nociceptive map in S1 and also the greater amplitude of EEG noxious responses in infants compared to adults (*Fabrizi et al., 2016*).

## Pain and the developing S1 cortex

This study highlights the importance of understanding the development of touch and pain processing in the human infant brain. The widespread S1 nociceptive topography discovered here implies that the infant S1 cortex would be unable to accurately localise noxious events and may lack the computational ability to reliably send noxious information to higher brain centres (*Thivierge and Marcus, 2007*; *Harding-Forrester and Feldman, 2018*). Heel lance is one of many skin-breaking procedures commonly performed in neonatal hospital care (*Laudiano-Dray et al., 2020*) and this study reveals the extent of cortical activation that follows just one such noxious procedure in the newborn. This contrasts with innocuous mechanical stimulation, such as touch, which activates a spatially restricted and somatotopically defined cortical area. Increasing evidence that repeated noxious experiences have adverse effects upon the developing brain (*Ranger and Grunau, 2014*; *Duerden et al., 2018*) underlines the importance of these results and the need for a better understanding of the mechanisms underlying the maturation of cortical nociceptive topographic maps.

## Materials and methods
### Participants

Thirty-two infants (35–42 gestational weeks at birth, 0–7 days of age, 12 female; *Table 1*) were recruited from the postnatal, special care, and high dependency wards within the neonatal unit at University College London Hospital. Infants received either (1) innocuous mechanical stimulation (touch) of the heel, (2) innocuous mechanical stimulation of the hand, or (3) a noxious mechanical stimulation (clinically required lance) of the heel. Six infants received touch stimulation of both the heel and hand. This sample size is sufficient as similar high impact works using single trial noxious stimulation or multiple mechanical stimulations have yielded significant results with group sample sizes of 5–15 (*Bartocci et al., 2006*) and 10–15 (*Arichi et al., 2012*), respectively. Ethical approval for this study was given by the NHS Health Research Authority (London – Surrey Borders) and the study conformed to the standards set by the Declaration of Helsinki. Informed written parental consent was obtained before each study.

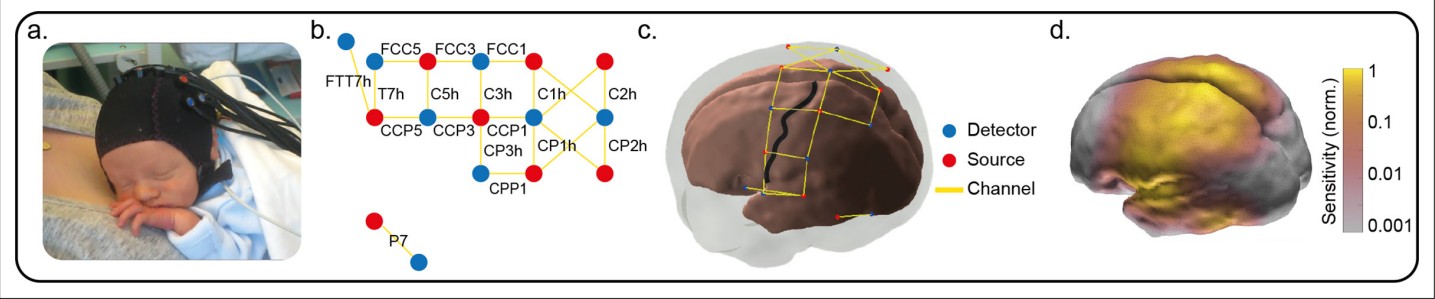

**Figure 4.** Optode locations and sensitivity map. (**a**) Typical functional near-infrared spectroscopy (fNIRS) setup on a neonate of 35$^{+2}$ weeks' gestational age (GA), 7 days' postnatal age (PNA). (**b**) Channel locations according to the international 10–5 placement system (*Oostenveld and Praamstra, 2001*). Names of four diagonal channels (channels overlaying Cz and CPz) are not included as these were omitted from the analysis due to low signal-to-noise ratio. (**c**) Locations of the fNIRS sources, detectors and resulting measurement channels registered to a 39-week anatomical atlas. The central sulcus has been highlighted with a black line and the location of S1 is the gyrus posterior to this sulcus. (**d**) Normalised fNIRS sensitivity illustrating the spatial coverage provided by the channel arrangement in panel (**c**). This sensitivity map was calculated using the photon measurement density functions derived from the TOAST++ light transport modelling package.

## Experimental design

Brain activity (fNIRS) was recorded following a noxious (clinically required heel lance) or innocuous (touch) mechanical stimulation of the limbs at the bedside in the neonatal unit.

## fNIRS recording

Infants wore a 21-channel array consisting of eight sources and eight detectors with inter-optode distances of 2.5–4 cm. The array was secured over the pericentral area of the scalp on the side contra-lateral to the stimulation with a custom-designed textile cap (EasyCap, *Figure 4a*). The infants' head circumference, ear to ear lateral semi-circumference, and nasion to inion distance were measured and the cap was placed on the head by aligning specific 10/5 locations (Cz, T7). This optode arrangement provided sensitivity coverage for the somatomotor cortex contralateral to the stimulation site and of the medial part on the ipsilateral side (*Figure 4d*). One source-detector pair was placed at a more ventral posterior location of the scalp (P7 of the international 10/5 positioning system) (*Figure 4b*). This channel was sensitive to the posterior temporal lobe and worked as a control channel (*Figure 4b–d*). A continuous wave recording system was used with two wavelengths of source light at 780 and 850 nm and a sampling rate of 10 Hz to measure changes in oxy- and deoxy-haemoglobin concentration (Gowerlabs NTS fNIRS system).

## Noxious mechanical stimulation

The noxious stimulus was a clinically required heel lance for blood sampling. Blade release was time-locked to the NIRS recording using an accelerometer attached to the lancet (*Worley et al., 2012*). The lancet was placed against the heel for at least 30 s prior to the release of the blade. This was to obtain a baseline period free from other stimulation. The heel was then squeezed 30 s after the release of the blade, again to ensure a post-stimulus period free from other stimuli. All lances were performed by the same trained nurse (MPL-D) using a disposable lancet, and standard hospital practice was followed at all times.

All infants were prone against their mother's chest. The mother, who was inclined on a chair or bed, was instructed to avoid moving or stimulating the infant during the 1 min before and after the release of the lance.

## Innocuous mechanical stimulation

Innocuous mechanical stimulation was delivered by light touch on the lateral edge of the infants' palms and/or heels using a hand-held tendon hammer (ADInstruments). A piezo-electric sensor mounted on the hammer head provided a synchronising signal to the NIRS recording. A train of up to 15 touches (average = 11.1) was delivered to each limb with a variable inter-stimulus interval of 35–60 s. This resulted in an average of 11.5 heel touches (range = 7–15) and 10.6 hand touches (range = 6–13) per

infant. Stimulus repetition did not cause habituation of the response to touch (no significant difference in the average time-series between the first and last 50% of the trials).

## Infant movement

To limit cortical activation not related to the investigated sensory stimuli, body movements were minimised during the lance procedure, as infants were swaddled (wrapped securely in clothes/blankets) against the mother's chest and the research nurse was holding the exposed foot throughout the period before and after the stimulus.

**Table 2.** Infant movements.

The number of infants who displayed movements or received tactile stimulation from their mother each second in the 30 s following lance.

| | Number of infant and maternal movements | | | | | | |
|---|---|---|---|---|---|---|---|
| Post-lance (s) | Hand | Head | Face | Foot | Arm | Mother touching face | Mother touching head |
| 1 | 1 | 1 | 4 | 0 | 2 | 1 | 1 |
| 2 | 1 | 1 | 6 | 1 | 2 | 1 | 1 |
| 3 | 1 | 0 | 6 | 1 | 2 | 1 | 1 |
| 4 | 1 | 0 | 6 | 0 | 2 | 1 | 1 |
| 5 | 1 | 1 | 6 | 0 | 2 | 1 | 1 |
| 6 | 1 | 1 | 6 | 0 | 2 | 1 | 0 |
| 7 | 0 | 1 | 4 | 0 | 1 | 1 | 0 |
| 8 | 0 | 1 | 3 | 0 | 1 | 1 | 0 |
| 9 | 0 | 1 | 2 | 0 | 1 | 1 | 0 |
| 10 | 0 | 0 | 1 | 0 | 1 | 1 | 0 |
| 11 | 0 | 0 | 1 | 0 | 1 | 1 | 0 |
| 12 | 0 | 0 | 1 | 0 | 1 | 1 | 0 |
| 13 | 0 | 0 | 1 | 0 | 1 | 1 | 0 |
| 14 | 0 | 0 | 1 | 0 | 1 | 1 | 0 |
| 15 | 0 | 0 | 1 | 0 | 0 | 1 | 0 |
| 16 | 0 | 0 | 1 | 0 | 0 | 1 | 0 |
| 17 | 0 | 0 | 0 | 0 | 0 | 1 | 0 |
| 18 | 0 | 0 | 0 | 0 | 0 | 1 | 0 |
| 19 | 0 | 0 | 0 | 0 | 0 | 1 | 0 |
| 20 | 0 | 0 | 0 | 0 | 0 | 1 | 0 |
| 21 | 0 | 0 | 0 | 0 | 0 | 1 | 0 |
| 22 | 0 | 0 | 0 | 0 | 0 | 1 | 0 |
| 23 | 0 | 0 | 0 | 0 | 0 | 1 | 0 |
| 24 | 0 | 0 | 0 | 0 | 0 | 1 | 0 |
| 25 | 0 | 0 | 0 | 0 | 0 | 1 | 0 |
| 26 | 0 | 0 | 0 | 0 | 0 | 1 | 0 |
| 27 | 0 | 0 | 0 | 0 | 0 | 1 | 0 |
| 28 | 0 | 0 | 0 | 0 | 0 | 1 | 0 |
| 29 | 0 | 0 | 0 | 0 | 0 | 1 | 0 |
| 30 | 0 | 0 | 0 | 0 | 0 | 1 | 0 |

Each movement or stimulation was scored as present (1) or not present (0) per infant, and the value in each cell represents the total number of infants (out of 11) for whom each movement or stimulation was observed.

Infant movements or maternal handling were recorded on video, which was synchronised with the NIRS recording using an LED light within the frame that was activated by the release of the lance (*Worley et al., 2012*) or annotated at the time of study using a stopwatch. For lance trials, movements or maternal handling were separated into body parts and coded per second as either 0 (not present) or 1 (present) for the 30 s post-stimulus (*Table 2*). Following the lance, two babies did not move, five babies made small movements (including small or brief grimace, head nod, twitch, small hand movement), four babies made larger movements (including arms, large or prolonged grimace, nod of head), and two babies received tactile stimulation from the mother (including positioning the head, stroking the face). Following touch, body movements were present in an average of 2.9 trials in 8/16 infants (heel) and 1.4 trials in 11/11 infants (hand). Although every effort was made to minimise trials affected by body movements, these were sometimes unavoidable following a single lance trial and therefore were included for both lance and touch analysis. However, this is unlikely to have affected our results because body movements were not systematically associated to the stimulus of interest as they spanned different body parts and latencies such that any associated cortical response would be cancelled during the averaging process. Indeed, comparison of the time-series following heel and hand touch with and without removing trials with known movement showed no significant difference (*Figure 1—figure supplement 2*).

## Data pre-processing

All data were pre-processed in Homer2 (*Huppert and Boas, 2005*). The four channels crossing over the midline had consistent poor signal quality (light intensity <0.01, SNR <2) and were not considered further (*Figure 4b and c*). Recordings from 1 to 8 (average: 0.74) channels in 11 (29%) trials were also removed for lance and touch stimuli. Data were then converted into optical density, motion artefacts were detected (change in amplitude >0.7 and/or change in standard deviation >15 over a 1 s time period) and then corrected using wavelet filtering (*Molavi and Dumont, 2012*). Instrumental drift and cardiac artefact were removed with a 0.01–0.5 Hz bandpass filter. Optical density changes recorded from all channels (likely related to stimulus-dependent systemic physiological changes) were removed using PCA [*Kozberg and Hillman, 2016*; *Tachtsidis and Scholkmann, 2016*]; one component removed for every subject. Finally, data were converted into changes in oxy- and deoxy-haemoglobin concentration ($\Delta$[HbO] and $\Delta$[Hb]) using the modified Beer–Lambert law (*Delpy et al., 1988*) with a differential path length factor of 4.39 (*Wyatt et al., 1990*). The continuous signal was then epoched from –5 to 30 s around the noxious and somatosensory stimuli. Somatosensory stimuli were averaged for each subject.

## Signal to noise

The signal-to-noise ratio (SNR) for lance and for touch was calculated. Despite the peak signal for lance being larger, the estimated SNR of the peak was lower, because more touch trials were averaged in this study: $\frac{(Peak_{Lance} * \sqrt{N_{Lance}})}{(Peak_{Touch} * \sqrt{N_{Touch}})}$ = ratio of lance to touch SNR; $\frac{\left(0.96 * \sqrt{11}\right)}{\left(0.35 * \sqrt{184}\right)}$ = 0.67.

## Channel-wise data analysis

Pre-processed data were then averaged across trials for each subject (except lance which only had one trial) and analysed using custom MATLAB scripts (Mathworks; version 16b). For each channel, significant changes in $\Delta$[HbO] and $\Delta$[Hb] were identified with a two-tailed t-test ($\alpha$ = 0.01) comparing each time point post-stimulus against the baseline. This baseline distribution was calculated as the mean of the individual baselines (–5 to 0 s before stimulus) according to:

$$\tfrac{1}{S} \sum_{i=1}^{S} x_i \, N\left(\tfrac{1}{S}\sum_{i=1}^{S}\mu_i, \tfrac{1}{S^2}\sum_{i=1}^{S}\sigma_i^2\right).$$

where $S$ is the number of subjects and $x_i \, N\left(\mu_i, \sigma_i^2\right)$ is the baseline for subject .

Bonferroni correction was used for multiple comparisons (17 channels × 300 samples = 5100 comparisons). Only changes (increases or decreases) which were continuously significant for at least 1 s (10% of the length of the post-lance period) were retained (*Guthrie and Buchwald, 1991*).

## Data analysis in image space

### Image reconstruction

The channel-wise data was used to create functional images using a cortically constrained linear reconstruction approach at each time point to obtain image time-series for each subject. The fNIRS array was registered to a 39-week gestational age anatomical mesh model with 784,391 nodes (*Brigadoi et al., 2014*) using tools from the AtlasViewer package (*Aasted et al., 2015*). Images were reconstructed using the DOT-HUB toolbox (*Cooper et al., 2021*) and the TOAST++ light transport modelling package (*Schweiger and Arridge, 2014a*, *Schweiger and Arridge, 2014b*), with zeroth-order Tikhonov regularisation with a regularisation hyperparameter of 0.1.

### Assessment of peak changes in Δ[HbO] and Δ[Hb]

Image space analysis was conducted separately from channel space. We first assessed if hand and foot touch and foot lance evoked a significant response in Δ[HbO] and Δ[Hb], by comparing their peak change to baseline. To do that, we averaged the image time-series and found the latency at which the maximum change occurred anywhere across the cortex. For each node, significant changes at that latency in Δ[HbO] and Δ[Hb] were assessed with a two-tailed t-test ($\alpha = 0.01$) comparing the 5 s window around the identified peak latency against the 5 s baseline. As in channel-wise analysis peak signal and baseline distributions were estimated by concatenating the individual baselines (−5 to 0 s pre-stimulus) and peak windows (−2.5 to 2.5 s around peak latency). Bonferroni correction was used for multiple comparisons (784,391 nodes = 784,391 comparisons). To display these results, we: (1) reconstructed an image using the average channel-wise data within the 5 s window around the peak latency (averaged in time) for hand touch, heel touch, and lance and (2) masked this image according to the result of the statistical test above.

### Comparison of peak changes in Δ[HbO] and Δ[Hb] between heel lance and touch

Next, we compared the significant changes in Δ[HbO] and Δ[Hb] between the heel lance and touch conditions in two ways: (1) comparing the magnitude of the response at peak latency at every node using a Student's t-test, (2) comparing the characteristics of the peak response (overall area of activation and latency, location, and spread of the peak change in Δ[HbO] and Δ[Hb]) using a non-parametric test.

Overall area of activation was defined by starting at the node with the peak change and continually expanding to include neighbouring nodes that (1) were connected by three face edges and (2) had a significant change from baseline. Difference in peak location was the Euclidean distance between the peaks. Spread of the peak was defined as the part of the overall area of activation around the peak where changes in Δ[HbO] (or Δ[Hb]) were at least half of the peak change (full-width half-maximum, FWHM).

The non-parametric null distribution was derived by calculating these differences between randomly selected sets of surrogate image time-series (bootstrapping on surrogate data). We here describe how we obtained surrogate image time-series and then how we conducted bootstrapping.

Each individual recording (i.e image time-series) can be considered as the linear sum of a signal of interest (i.e. the response to the stimulus) and a stationary random noise component. The assumption is that the signal is the same in each recording while the noise changes (*van Drongelen, 2007*). Therefore, if we were to conduct another recording on another subject, the new data would be the linear sum of the *same* signal that we find in the original data but *different* random noise. Creating surrogate data consists in generating new random noise to add to the signal estimated from our data. To do this we: (1) estimate the signal by averaging across individual recordings (i.e. subjects) in response to the same stimulus modality; (2) isolate the noise in our data by subtracting this estimate from each recording; (3) *phase-randomise* each noise time-series. Phase-randomisation is applied independently to each node time-series in the frequency domain. This means that the phase component of the complex-valued signal is rotated at each frequency by an independent random variable chosen from the uniformly distributed range of 0 and 2π (*Theiler et al., 1992*). At the end of this process, we have a new set of surrogate noise time-series.

To generate the full non-parametric null distribution against which to compare our data, we used bootstrapping. To estimate each sample of the null distribution, we calculated the differences in

overall area of activation and peak amplitude, latency, position, and FWHM between two random sets of surrogate data without any systematic difference. To create the random sets, we: (1) pooled together all the newly obtained surrogate noise time-series; (2) added the grand average (across lance and touch) signal (as we do not want systematic differences between sets to estimate a null distribution); (3) randomly split (with repetition) these surrogate data into two sets. We repeated this 1000 times in order to obtain the full non-parametric null distribution (bootstrapping). An experimental difference outside the 95% confidence interval was considered significant ($p < 0.05$).

## Acknowledgements

This work was funded by the Medical Research Council UK (MR/M006468/1, MR/L019248/1, and MR/S003207/1). RJC is funded by EPSRC Fellowship EP/N025946/1.

## Additional information

### Funding

| Funder | Grant reference number | Author |
|---|---|---|
| Medical Research Council | MR/M006468/1 | Judith Meek<br>Lorenzo Fabrizi<br>Maria Fitzgerald |
| Medical Research Council | MR/L019248/1 | Lorenzo Fabrizi |
| Engineering and Physical Sciences Research Council | EP/N025946/1 | Robert J Cooper |
| Medical Research Council | MR/S003207/1 | Judith Meek<br>Lorenzo Fabrizi<br>Maria Fitzgerald |

The funders had no role in study design, data collection and interpretation, or the decision to submit the work for publication.

### Author contributions

Laura Jones, Data curation, Formal analysis, Investigation, Methodology, Project administration, Writing - original draft, Writing - review and editing; Madeleine Verriotis, Formal analysis, Investigation, Methodology; Robert J Cooper, Formal analysis, Methodology, Resources, Validation; Maria Pureza Laudiano-Dray, Investigation, Supervision; Mohammed Rupawala, Formal analysis; Judith Meek, Clinical supervision, Conceptualization, Formal analysis, Funding acquisition, Methodology, Supervision, Writing - review and editing; Lorenzo Fabrizi, Formal analysis, Funding acquisition, Investigation, Methodology, Writing - review and editing; Maria Fitzgerald, Conceptualization, Funding acquisition, Methodology, Project administration, Supervision, Writing - original draft, Writing - review and editing

### Author ORCIDs

Laura Jones http://orcid.org/0000-0001-5755-4977
Madeleine Verriotis http://orcid.org/0000-0003-3019-0370
Lorenzo Fabrizi http://orcid.org/0000-0002-9582-0727
Maria Fitzgerald http://orcid.org/0000-0003-4188-0123

### Ethics

Human subjects: Ethical approval for this study was given by the NHS Health Research Authority (London - Surrey Borders) and the study conformed to the standards set by the Declaration of Helsinki. Informed written parental consent was obtained before each study (REC no: 11/LO/0350; NIHR Portfolio Study ID: 12036). Separate media consent was obtained from the parent to use a photo of their child in academic publications (Figure 4a).

### Decision letter and Author response

Decision letter https://doi.org/10.7554/eLife.71655.sa1

Author response https://doi.org/10.7554/eLife.71655.sa2

## Additional files

### Supplementary files
• Transparent reporting form

### Data availability
All raw data files are open access and are available to download from Figshare (https://doi.org/10.6084/m9.figshare.13252388.v3).

The following dataset was generated:

| Author(s) | Year | Dataset title | Dataset URL | Database and Identifier |
|---|---|---|---|---|
| Jones L, Verriotis M, Cooper RJ, Laudiano-Dray M, Rupawala M, Meek M, Fabrizi L, Fitzgerald M | 2021 | Widespread nociceptive maps in the human neonatal somatosensory cortex | https://doi.org/10.6084/m9.figshare.13252388.v3 | figshare, 10.6084/m9.figshare.13252388.v3 |

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
