## [Editor Report]

This paper is of interest to developmental neuroscientists who study the early stages of cortical maturation and specialization, particularly in the context of somatosensory and pain system development. The authors suggest that, relative to the infant touch somatotopic map, the infant nociceptive map is more widespread and poorly localised, consistent with infants' poorly directed pain behaviour.

---

## [Decision Letter]

**Decision letter after peer review:**

Thank you for submitting your article "Widespread nociceptive maps in the human neonatal somatosensory cortex" for consideration by *eLife*. Your article has been reviewed by 2 peer reviewers, and the evaluation has been overseen by Chris Baker as Reviewing and Senior Editor. The reviewers have opted to remain anonymous.

Essential revisions:

Please fully address the concerns and suggestions of the reviewers as laid out in their reviews below. Specifically:

1) Address the statistical concerns raised by both reviewers with new analyses. In particular, this will require including direct statistical testing of the two maps (R1) as well as addressing concerns around circularity (R1) and the different numbers of trials for each condition (R2).

2) Reconsider use of the term "widespread" to describe responses to the painful stimuli (R1 and R2). Terms such as "mislocalized" or "disorganized" may be more appropriate.

3) Treat motion equivalently for the two conditions in the analyses (R2).

*Reviewer #1 (Recommendations for the authors):*

1. Differences in regions of activations constituting touch maps and nociceptive maps are never actually statistically tested:

See the public review for an outline of the issue. The suggested solution is to directly compare the noxious and touch evoked activations and demonstrate statistically significant differences between them.

2. Nociceptive maps are equally as widespread as touch maps:

See the public review for an outline of the issue. The distinction between "mislocalisation" versus "widespread" needs to be clarified, as the claim that nociceptive maps are widespread is ubiquitous and currently not sufficiently contextualised relative to the touch maps. Or at least the apparent tension between this core claim and the lack of difference in area of activation for the noxious and non-noxious stimulus needs explicit discussion. Of course, the "mislocalisation" claim would have to be backed up by a demonstration of statistically significant differences between the maps, which is currently not provided (see comment 1 above).

3. The analyses that produce areas of activation in source space (image space) are circular:

See the public review for an outline of the issue. The core message of this manuscript is that infants' nociceptive maps are more poorly localised than touch maps. This could be demonstrated in sensor space without any need to do source reconstruction. As outlined in comment 1 above, a direct statistical comparison of noxious evoked and touch evoked activity in sensor space would demonstrate the core message the authors wish to convey, and does not need source reconstruction meaning the circular analysis can be omitted altogether. And as mentioned above, all statistically significant results found in source space are not novel, so very little would be lost in keeping the analysis and results in sensor space and simple establishing statistically significant differences between the nociceptive and touch maps.

4. Limitations of the fNRIS technique and of the link between properties of nociceptive maps (widespread/mislocalisation) and poorly directed behaviour are not adequately discussed:

See the public review for an outline of the issue. This major limitation of the fNIRS technique requires much more serious and explicit discussion.

5. Linking comments 1, 2, and 4 to the initial hypothesis:

The hypothesis stated in the introduction is as follows: "We hypothesised that this developmental process is reflected in ascending nociceptive signals to SI, leading to widespread cortical activation and poor spatial localisation of noxious events in early life". The assessment of the poor localisation and widespread activation of noxious evoked activity is assessed and explicitly interpreted relative to touch evoked activity: "Noxious stimulation of the heel in the newborn evokes a more widespread cortical activation than innocuous stimulation, that extends into inferior regions of S1, normally associated with representation of the hand.".

However, using statistical tests to directly compare the noxious evoked activity to the touch evoked activity, there was neither a statistically significant difference between the noxious and touch peak localisations (i.e. peak locations do not differ) nor their extent of cortical activations (i.e. areas of activations do not differ). This subset of results seem to directly go against the initial hypothesis and interpreted conclusions.

A much clearer link between the tests run, results provided, and hypothesis put forward needs to be outlined, as it is currently very unclear why some results (e.g. activations assessed relative to baseline) should be considered supportive of the initial hypothesis but not indicative of fNIRS-related haemodynamic confounds (e.g. non-neural cardiovascular response signals such as tissue-damage evoked blood pressure changes), while other results (e.g. direct statistical comparisons between noxious and touch evoked activations) that seem to contradict the initial hypothesis are not considered as such by the authors.

*Reviewer #2 (Recommendations for the authors):*

In order to address the issue of repetition suppression in the touch condition, it would be of interest to see either an analysis of change in the touch responses across trials, or a comparison between touch and nociceptive trials based on only the first touch trial.

There is some small irregularity in the reporting of motion in the pain condition. The N for this condition in Table 1 is 11, but when reporting motion the authors state that 2 participants didn't move, 6 made small movements, and 4 made large movements.

Perhaps I am misreading the analysis procedure, but if the authors really are averaging responses to the touch/pain conditions and using a single, mean value for each time point in statistical tests (compared in a one-sample t-test to the baseline distribution?), then I do believe these analyses should be adjusted to retain variability across participants in response to the stimuli of interest.

I was also surprised by the information on Bonferroni corrections. The authors were really multiplying their p values by thousands (channel-wise analyses) or hundreds of thousands (image-based analyses) of comparisons and arriving at corrected p values less than p = .001, in tests with 11 infants and a single trial per infant? I would not expect this sort of effect size in infant fNIRS research. Am I misunderstanding how a Bonferroni correction would be applied here?

[Editors’ note: further revisions were suggested prior to acceptance, as described below.]

Thank you for resubmitting your work entitled "Widespread nociceptive maps in the human neonatal somatosensory cortex" for further consideration by *eLife*. Your revised article has been evaluated by Chris Baker (Senior Editor).

The manuscript has been improved but the reviewers raised some remaining concerns and made some good suggestions. There are some remaining issues that need to be addressed, as outlined below:

– Please show the S1 ROI as suggested by R1

– In the revision, you show that there is no discernible impact of motion on the heel touch trials in response to a concern raised by R2. However, this does not necessarily solve the issue raised. The suggestion to treat both trial types equally (as raised in the first round of review) should be followed.

– Both reviewers and myself had a hard time following exactly how the image space analyses are conducted and whether the concern about circularity has been addressed or not. It sounds as if different nodes or timepoints are being selected on the basis of the responses observed, which still has the potential to introduce a selection bias (i.e. circularity). Please clarify and expand the text in describing the analyses.

– As noted by R2, the text at the top of page 5 is confusing. More generally, the descriptions in the text tend to be quite dense and complicated. Please revise the section noted and consider simplifying the text throughout to make the descriptions of the analyses and the corresponding conclusions easier to follow.

– As suggested by R2, please consider providing the preprocessed data in addition to the raw files.

*Reviewer #1 (Recommendations for the authors):*

This manuscript has improved since initial submission, and the authors have appropriately addressed the points in my previous review. I have one final suggestion that I think would greatly help the reader and improve the manuscript further.

It would be very helpful to see an outline of the S1 area, as defined and used in this study. The brain surface onto which the activity has been projected during image reconstruction is very smooth (see Figures2-4); it is not easy to make out with any confidence where any of the key landmarks are e.g. central sulcus, precentral sulcus, postcentral sulcus. In all of your source space maps, only two brain orientations are used: view from above and view from behind. I would suggest adding an additional supplementary figure of these two brain orientations with the S1 area clearly delineated. Or alternatively, putting in a fourth panel into Figure 4 to delineate the area, as Figure 4 already contains quite a bit of localisation/orientation information (regarding optode setup), so seems a sensible place for defining the ROI of interest i.e. area S1.

Having this area clearly defined and detailing how it was defined (e.g. anatomically or functionally defined; manually or automatically outlined; use of an established functional or structural atlas, etc) will really help readers interpret the activity maps more accurately.

*Reviewer #2 (Recommendations for the authors):*

In general, the authors made clear and reasonable edits to their manuscript that addressed many of the concerns with the original paper. I do, however, think there are a few points that still need attention:

– One of my original concerns was that motion-contaminated trials were dropped in the heel touch condition but not the heel lance condition. In their revision, the authors compare heel touch data with and without motion-contaminated trials and find no significant differences, but I'm not sure this is the right way to deal with this concern; instead I still think motion should be treated the same across the two conditions. Large artifacts introduced by motion could fail to create systematic differences in heel touch data with and without motion, but still account for apparent differences between lance and touch data. If motion artifacts aren't a significant concern to include in the lance condition because they average one another out, that approach ought to be fine for the heel touch condition as well. If artifacts make the heel touch data too noisy, despite the greater number of trials, then their presence in the single-trial lance data ought to be treated as a serious source of contamination in the lance data and excluded from both.

– The authors dealt with Reviewer 1's concern about circularity in the image space analysis by using the image space peak, rather than the source space peak, to select the time period for statistical testing. If anything, I think this makes the circularity problem worse. Selecting a subset of the data based on one feature (proximity to peak concentration values), and then using those data in a subsequent statistical test to characterize the functional properties of the cortical location in question is the issue; the data used to select need to be independent of the data used to test. This seems like a tricky problem to solve here, but maybe a leave-one-subject-out procedure, in which the authors iterate through leaving out one subject when selecting the peak, extract response data from that subject, and then move onto the next iteration in order to obtain their test data would help? (Though I agree with Reviewer 1's original point that there's nothing wrong with just analyzing the data in source space, and omitting the image space analysis entirely).

– I was confused by the new text at the top of page 5, which (as far as I can tell) states that at every location where there was a significant effect of lance > baseline, there was also a significant lance > heel touch effect; and conversely that at every location where touch > baseline, there was also a touch > lance effect. Given that there was overlap in the effects of lance > baseline and touch > baseline, how could this possibly be true? Maybe this is not what the authors were trying to say, and they only meant that there are some regions in which there is a significant lance> touch effect and others in which there is a significant touch>lance effect (as shown in Figure 3c). If that's the case, I would suggest rewording this new text, as it was difficult to parse.

– I went to check the data files, and it seems that the authors only provide the raw.nirs files and demographic characteristics in the repository. Reconstructing their preprocessing stream would be extremely time consuming; I suggest they should also provide the data in the form used for statistical tests.

---

## [Author Response]

Essential revisions:Please fully address the concerns and suggestions of the reviewers as laid out in their reviews below. Specifically:1) Address the statistical concerns raised by both reviewers with new analyses. In particular, this will require including direct statistical testing of the two maps (R1) as well as addressing concerns around circularity (R1) and the different numbers of trials for each condition (R2).

We have now added a direct comparison of the HbO response amplitude between lance and heel touch at each node. The results have been added to Figure 3 and are discussed in the relevant methods and Results sections.

We have removed the latency restriction from the source-space analysis (i.e. using the peak latency identified from the channel-space) and updated the results appropriately; the results have not changed as a consequence of this.

We have now included a brief description of analysis which demonstrates that there was no significant habituation following the repeated heel touch stimulation. We have provided a figure of the resulting time-series in our response to reviewers which can be added as a supplementary image if required.

2) Reconsider use of the term "widespread" to describe responses to the painful stimuli (R1 and R2). Terms such as "mislocalized" or "disorganized" may be more appropriate.

We have been careful throughout the manuscript to only refer to a ‘widespread’ response specifically within the S1 cortex (and equally refer to the touch response being localised within the S1) and we feel the term ‘widespread’ is more easily understood than ‘mislocalised’. Although the foot touch and lance HbO responses have the same size area of activation, the response to foot touch is localised within the S1, but extends beyond this into the motor cortex. However, the response to lance does not extend into the same area of the motor cortex, and instead extends into the entire S1. Thus, although the two maps are misaligned, the more important point here is that nociceptive maps are widespread in terms of S1 activation. We have made small clarifications throughout the manuscript to ensure there is no confusion regarding our use of the word widespread.

3) Treat motion equivalently for the two conditions in the analyses (R2).

We have performed additional analyses comparing the heel touch data with and without removing the trials with known movement. A sentence summarising this analysis and the results have been added to the methods section. No significant differences were found, suggesting that this difference between the heel touch and lance trials did not contribute significantly to the present findings. We have included a figure of the resulting channel-space time-series in our response to the reviewers and this can be added as a supplementary figure if needed.

Reviewer #1 (Recommendations for the authors):1. Differences in regions of activations constituting touch maps and nociceptive maps are never actually statistically tested:See the public review for an outline of the issue. The suggested solution is to directly compare the noxious and touch evoked activations and demonstrate statistically significant differences between them.

We have now included this analysis in the manuscript: ‘In regions significantly active following the lance stimulus, direct comparison at each node shows that the response following the heel lance was indeed larger than the response following the heel touch, conversely, the response following the heel lance was significantly smaller in the region only active following the heel touch’. We have also added an image of these results to figure 3.

2. Nociceptive maps are equally as widespread as touch maps:See the public review for an outline of the issue. The distinction between "mislocalisation" versus "widespread" needs to be clarified, as the claim that nociceptive maps are widespread is ubiquitous and currently not sufficiently contextualised relative to the touch maps. Or at least the apparent tension between this core claim and the lack of difference in area of activation for the noxious and non-noxious stimulus needs explicit discussion. Of course, the "mislocalisation" claim would have to be backed up by a demonstration of statistically significant differences between the maps, which is currently not provided (see comment 1 above).

We have been careful throughout the manuscript to only refer to a ‘widespread’ or ‘localised’ response specifically within the S1 cortex. Although the heel touch and lance HbO responses have the same size area of activation, the response to heel touch is localised within the S1 (but extends beyond this into the motor cortex), and the heel lance response is widespread within the S1. The spread within the S1 is the important point here, rather than the size of the overall responses. We have also added the requested analysis to help further this point (see above) and have made small edits to clarify throughout the manuscript.

3. The analyses that produce areas of activation in source space (image space) are circular:See the public review for an outline of the issue. The core message of this manuscript is that infants' nociceptive maps are more poorly localised than touch maps. This could be demonstrated in sensor space without any need to do source reconstruction. As outlined in comment 1 above, a direct statistical comparison of noxious evoked and touch evoked activity in sensor space would demonstrate the core message the authors wish to convey, and does not need source reconstruction meaning the circular analysis can be omitted altogether. And as mentioned above, all statistically significant results found in source space are not novel, so very little would be lost in keeping the analysis and results in sensor space and simple establishing statistically significant differences between the nociceptive and touch maps.

We have removed the latency restriction from the source-space analysis (i.e. we find the peak response in source-space rather than use the peak latency identified from the channel-space) and updated the results appropriately; the results have not changed as a consequence of this.

4. Limitations of the fNRIS technique and of the link between properties of nociceptive maps (widespread/mislocalisation) and poorly directed behaviour are not adequately discussed:See the public review for an outline of the issue. This major limitation of the fNIRS technique requires much more serious and explicit discussion.

It is possible that systemic responses due to an increase in blood pressure may still be an issue following a lance stimulus. However, we believe that this is not a contributing factor in our results. If there was any remaining systemic response across the scalp following the heel lance, then we would expect all nodes to have a larger response following the lance compared to heel touch, or at least all channels/nodes to show a significant increase from baseline. However, despite the majority of the S1 (Figures 2c and 3a) being significantly active following the heel lance, there are still 9/17 channels that do not have a significant increase in HbO from baseline, and a region of the cortex which is more active following heel touch stimulation.

Moreover, although the control channel had a significant increase from baseline following the heel lance, this is not evidence that the heel lance elicited widespread systemic changes, as the heel touch also elicited a response in the control channel.

The overall areas of activation are the same for both stimuli, but they are not aligned. Although we use the term ‘widespread’, we are careful to only say as such within the S1, and not widespread across the whole cortex. Only a widespread response across the whole cortex could be viewed as being confounded by systemic changes. We have now added the discussion of systemic changes to the manuscript.

5. Linking comments 1, 2, and 4 to the initial hypothesis:The hypothesis stated in the introduction is as follows: "We hypothesised that this developmental process is reflected in ascending nociceptive signals to SI, leading to widespread cortical activation and poor spatial localisation of noxious events in early life". The assessment of the poor localisation and widespread activation of noxious evoked activity is assessed and explicitly interpreted relative to touch evoked activity: "Noxious stimulation of the heel in the newborn evokes a more widespread cortical activation than innocuous stimulation, that extends into inferior regions of S1, normally associated with representation of the hand.".However, using statistical tests to directly compare the noxious evoked activity to the touch evoked activity, there was neither a statistically significant difference between the noxious and touch peak localisations (i.e. peak locations do not differ) nor their extent of cortical activations (i.e. areas of activations do not differ). This subset of results seem to directly go against the initial hypothesis and interpreted conclusions.A much clearer link between the tests run, results provided, and hypothesis put forward needs to be outlined, as it is currently very unclear why some results (e.g. activations assessed relative to baseline) should be considered supportive of the initial hypothesis but not indicative of fNIRS-related haemodynamic confounds (e.g. non-neural cardiovascular response signals such as tissue-damage evoked blood pressure changes), while other results (e.g. direct statistical comparisons between noxious and touch evoked activations) that seem to contradict the initial hypothesis are not considered as such by the authors.

Please see previous responses.

We have now added the discussion of systemic changes to the manuscript and altered the hypothesis to state that we expected widespread activation within the S1 specifically.

Reviewer #2 (Recommendations for the authors):In order to address the issue of repetition suppression in the touch condition, it would be of interest to see either an analysis of change in the touch responses across trials, or a comparison between touch and nociceptive trials based on only the first touch trial.

In order to address this, we have carried out a statistical comparison of the first and last 50% of foot touch trials at each sample post-lance across all channels. Similar comparison was made between heel touch data with and without removing trials with known movement. We did not find any significant differences for either comparison, and the average time series can be seen in Author response image 1. Solid lines represent the average and dashed lines represent the standard deviation. We have now added these results to the manuscript and the image can be included as a supplementary if required.

**Author response image 1. sa2fig1:** 

There is some small irregularity in the reporting of motion in the pain condition. The N for this condition in Table 1 is 11, but when reporting motion the authors state that 2 participants didn't move, 6 made small movements, and 4 made large movements.

Thank you for noting this error. This should be 5 made small movements.

Perhaps I am misreading the analysis procedure, but if the authors really are averaging responses to the touch/pain conditions and using a single, mean value for each time point in statistical tests (compared in a one-sample t-test to the baseline distribution?), then I do believe these analyses should be adjusted to retain variability across participants in response to the stimuli of interest.

We apologise for the confusion as this was an error in the manuscript; we averaged across trials not subjects. We have changed the relevant section in the methods.

‘Pre-processed data were then averaged across trials (except lance which only had one trial) and analysed using custom MATLAB scripts (Mathworks; version 16b).’

I was also surprised by the information on Bonferroni corrections. The authors were really multiplying their p values by thousands (channel-wise analyses) or hundreds of thousands (image-based analyses) of comparisons and arriving at corrected p values less than p = .001, in tests with 11 infants and a single trial per infant? I would not expect this sort of effect size in infant fNIRS research. Am I misunderstanding how a Bonferroni correction would be applied here?

The p-values provided in Figure 1 – source data 1 are indeed the corrected p-values, which were calculated by multiplying each p value (one per sample post-lance) by the number of comparisons (5100). Within the periods of significant change compared to baseline, we have provided the minimum and maximum p value.

[Editors’ note: further revisions were suggested prior to acceptance, as described below.]

The manuscript has been improved but the reviewers raised some remaining concerns and made some good suggestions. There are some remaining issues that need to be addressed, as outlined below:– Please show the S1 ROI as suggested by R1

We have highlighted the central sulcus to the brain in Figure 4 so that it is now clear which optodes were sensitive to the postcentral gyrus and so used to delineate the somatosensory cortex. We have also modified the key in Figures 1 (channel space analysis) to include the 10-5 labels so the relationship between channel space and cortical surface is clearer.

– In the revision, you show that there is no discernible impact of motion on the heel touch trials in response to a concern raised by R2. However, this does not necessarily solve the issue raised. The suggestion to treat both trial types equally (as raised in the first round of review) should be followed.

This has now been done and the new results reported in the manuscript. Treating both trial types equally had increased the significance of the data and this is discussed in lines 220-224. The impact of motion is also added to the Methods (lines 430-439)

– Both reviewers and myself had a hard time following exactly how the image space analyses are conducted and whether the concern about circularity has been addressed or not. It sounds as if different nodes or timepoints are being selected on the basis of the responses observed, which still has the potential to introduce a selection bias (i.e. circularity). Please clarify and expand the text in describing the analyses.

The initial concern arose because we originally selected time windows to analyse in image space from results from channel space analysis. To address this concern about circularity we have separated channel and image space analysis completely. In image space we assess the significance of peak change against baseline. Importantly the peak change was searched across ANY node and latency avoiding any bias. We clarified this section of the Methods (lines 487-488 and 495-519).

– As noted by R2, the text at the top of page 5 is confusing. More generally, the descriptions in the text tend to be quite dense and complicated. Please revise the section noted and consider simplifying the text throughout to make the descriptions of the analyses and the corresponding conclusions easier to follow.

This section has been rewritten and the analysis described in simpler terms (lines 158-173)

– As suggested by R2, please consider providing the preprocessed data in addition to the raw files.

The preprocessed data will be available upon reasonable request to Dr Laura Jones, laura.a.jones@ucl.ac.uk.data